# Characterization of Short Chain Fatty Acids Produced by Selected Potential Probiotic *Lactobacillus* Strains

**DOI:** 10.3390/biom12121829

**Published:** 2022-12-07

**Authors:** Suchera Thananimit, Nuntiya Pahumunto, Rawee Teanpaisan

**Affiliations:** 1Division of Biological Science, Faculty of Science, Prince of Songkla University, Songkhla 90110, Thailand; 2Center for Genomics and Bioinformatics Research, Faculty of Science, Prince of Songkla University, Songkhla 90110, Thailand; 3Department of Oral Diagnostic Sciences, Faculty of Dentistry, Prince of Songkla University, Songkhla 90110, Thailand

**Keywords:** probiotics, *Lacticaseibacillus paracasei* SD1, *Lacticaseibacillus rhamnosus* SD11, butyrate, anti-colon cancer, short-chain fatty acids (SCFAs)

## Abstract

Short-chain fatty acids (SCFAs), particularly butyrate, have received considerable attention with regard to their anti-cancer efficacy in delaying or preventing colorectal cancer. Several studies have reported that certain probiotic strains could produce SCFAs; however, different strains yielded different amounts of SCFAs. This study explored the ability to produce SCFAs of the following probiotic strains: *Lacticaseibacillus paracasei* SD1, *Lacticaseibacillus rhamnosus* SD4, *Lacticaseibacillus rhamnosus* SD11, and *Lacticaseibacillus rhamnosus* GG. *L. paracasei* SD1 and *L. rhamnosus* SD11 exhibited high butyrate production, particularly when the strains were combined. The functions of the SCFAs were further characterized; the SCFAs exerted a positive anti-cancer effect in the colon via various actions, including inhibiting the growth of the pathogens related to colon cancer, such as *Fusobacterium nucleatum* and *Porphyromonas gingivalis*; suppressing the growth of cancer cells; and stimulating the production of the anti-inflammatory cytokine IL-10 and antimicrobial peptides, especially human β-defensin-2. In addition, the SCFAs suppressed pathogen-stimulated pro-inflammatory cytokines, especially IL-8. The results of this study indicated that selected probiotic strains, particularly *L. paracasei* SD1 in combination with *L. rhamnosus* SD11, may serve as good natural sources of bio-butyrate, which may be used as biotherapy for preventing or delaying the progression of colon cancer.

## 1. Introduction

The symbiotic interaction of microorganisms in the gut is related to the health and disease of humans and animals. Lifestyle-related disorders can disrupt the homeostasis of the microbiota, leading to an imbalance in the microflora known as dysbiosis, which can cause various disorders, such as allergic diseases, obesity, diabetes, and inflammatory bowel disease [1,2], including colorectal cancer (CRC) [3,4]. Patients with CRC have exhibited a marked reduction in butyrate-producing bacteria [5] and an increase in the pathogens associated with CRC, especially *Fusobacterium nucleatum* [6,7,8].

To overcome an imbalance in gut microbiota, probiotics have been proposed as an alternative solution for the prevention and treatment of CRC [9,10]. Probiotic supplements have been used as a biotherapeutic because of their anti-cancer properties when consumed in sufficient quantities [11,12]. One of the mechanisms behind the beneficial effects of probiotics involves their bioactive metabolites, especially short-chain fatty acids (SCFAs), which mainly comprise acetic, propionic, and butyric acids [13,14].

SCFAs have been reported to have positive effects on human health via different actions, such as by providing the main source of energy for colon cells and exerting anti-microbial and anti-inflammatory effects [15]. Among the SCFA metabolites, butyrate has been considered to play a key role in the prevention and treatment of colon cancer by improving the function of colon cells [16], promoting the cell-cycle arrest and cell apoptosis of cancer, and stimulating immunomodulation [17]. Previous reviews have reported that various probiotic strains, e.g., *Bifidobacterium*, *Lactobacillus*, *Clostridium butyricum*, etc., could provide a benefit in gastrointestinal cancer prevention and treatment as butyric acid producers [18,19]. However, different probiotic strains could produce different amounts of SCFAs [20]; this may depend on the type and the individual strains, as well as the temperature and incubation time. This information has not yet been elucidated; therefore, further research is needed to obtain this information.

In this study, the SCFA production and functions of four probiotic strains, *Lacticaseibacillus paracasei* SD1 (formerly *Lactobacillus paracasei* SD1), *Lacticaseibacillus rhamnosus* SD4 (formerly *Lactobacillus rhamnosus* SD4), *Lacticaseibacillus rhamnosus* SD11 (formerly *Lactobacillus rhamnosus* SD11), and the commercial probiotic *Lacticaseibacillus rhamnosus* GG (formerly *Lactobacillus rhamnosus* GG) are characterized. The strains were selected based on their properties as probiotic strains. *L. rhamnosus* SD4 showed the highest adhesion to human oral keratinocytes H357 [21]. *L. paracasei* SD1 and *L. rhamnosus* SD11 could produce antimicrobial effects against pathogens related to oral and enteric diseases [22,23]. Recently, such strains have shown potential in clinical trials to act as probiotics for reducing cariogenic pathogens and promoting caries prevention [24,25,26,27]. In addition, receiving *L. paracasei* SD1 and *L. rhamnosus* SD11 could stimulate immunity in the saliva [28]. *L. rhamnosus* GG is a well-known commercial probiotic strain that has been extensively studied for its probiotic properties, including the prevention of gastrointestinal cancer [19]. Therefore, by focusing on their SCFA production, such selected strains may have the potential for CRC prevention. The objectives of this study were to examine the SCFA production abilities of the selected probiotic strains and the functions of the SCFAs.

## 2. Material and Methods

### 2.1. Probiotic and Pathogenic Strains

The selected probiotic strains used in this study were *L. paracasei* SD1, *L. rhamnosus* SD4, and *L. rhamnosus* SD11, which were derived from our collection of the Department of Oral Diagnostic Sciences at the Faculty of Dentistry, Prince of Songkla University. The strains were collected from the saliva of healthy children without dental caries [29]. The strains were initially identified based on the typical morphology on an MRS medium, which is catalase-negative, Gram-positive, rod-shaped, and non-spore-forming, and the biochemical tests. Subsequently, all the strains were confirmed using molecular methods, including the restriction fragment length polymorphism analysis of PCR-amplified 16S RNA [30], 16S rRNA PCR-Denaturing gradient gel electrophoresis [31], and 16S rRNA sequencing. A commercial probiotic *L. rhamnosus* GG (ATCC 53103) was purchased from the American Type Culture Collection (Manassas, VA, USA).

Four pathogen strains, *Salmonella enterica*, enterotoxigenic *Escherichia coli* (ETEC), *Porphyromonas gingivalis* ATCC 33277, and *F. nucleatum* ATCC 25,586 were used in this study. All the strains were cultured in a brain heart infusion (BHI) broth for 24–48 h at 37 °C under the appropriate conditions for the individual strains.

### 2.2. Optimization of SCFAs Production and Conditions

To explore the optimal conditions for achieving the maximum SCFA production of individual strains, the experiments were set out at the different temperatures and incubation times for monitoring the bacterial growth and SCFA production. The strains were grown in an MRS broth for 12, 24, and 48 h at 30, 37, 40, and 45 °C, respectively. After incubation, the culture was centrifuged at 8500 rpm at 4 °C for 10 min, and 100 mL of supernatant was prepared as a powder by lyophilization. The lyophilized powder was then dissolved in 10 mL of ultrapure water and mixed with 4 g of NaCl, 2 mL of 50% H_2_SO_4_, and 5 mL of diethyl ether. The SCFAs were extracted by shaking for 30 min in a reaction vessel. After centrifugation at 5000 rpm for 10 min (4 °C), the upper diethyl ether fractions were collected and subsequently allowed to dry at 25 °C. Next, they were suspended in 200 µL of PBS and examined for protein concentration using the Bradford method before further analysis.

The quantity of SCFAs was quantitated using GC-MS analysis (Agilent 7890A GC system with a flame ionization detector, Agilent Technologies Inc., Santa Clara, CA, USA) and a GC column (Agilent J & W GC column DB-FATWAX UI; 30 m × 0.25 mm × 0.25 μm). The initial oven temperature was 100 °C for 2 min, raised to 220 °C by 15 °C/min, and then maintained at 220 °C for 1 min. The temperatures of the flame ionization detector and injection port were 250 and 240 °C, respectively. The injected sample for analysis was 1 μL, and the running time for the individual sample was 10 min. The flow rates of nitrogen, hydrogen, and synthetic air were set at 25, 30, and 300 mL/min, respectively. A calibration curve of butyric acid with a GC grade purity (Sigma Aldrich, St. Louis, MO, USA) ranging from 0.08 to 50 mM was used for quantification of the butyric acid with the Agilent ChemStation plus software.

### 2.3. Antimicrobial Activity of SCFAs against Pathogens

#### 2.3.1. Agar Well Diffusion Method

One hundred microliters of tested pathogenic strains in BHI broth, equivalent to 10^8^ CFU/mL, were mixed with 20 mL of warm melted BHI agar. The mixture was poured into a plate with a 6 mm diameter metal cup. After the solidifying of the BHI agar, the metal cups were removed, and the well was filled with 80 µL of SCFAs extracted from each probiotic strain. The antimicrobial activity of the SCFA extract was determined by measuring the diameter of the inhibition zone in millimeters. The experiments were repeated thrice.

#### 2.3.2. Broth Microdilution Method

Minimum inhibitory concentration (MIC) was performed using the broth microdilution method. Each SCFA extract (100 µL) was subjected to 2-fold serial dilutions with BHI broth. A 100 µL suspension of each pathogenic strain tested in BHI broth at 10^8^ CFU/mL was added to each well and incubated at 37 °C under suitable conditions. The BHI broth was used as the negative control. MIC was defined as the lowest concentration of the SCFA extracts that completely inhibited growth in comparison with the non-treated control. All experiments were repeated thrice.

#### 2.3.3. Checkerboard Assay

The SCFA extracts of the individual strains were prepared at four to two times the MIC, and 50 μL of one probiotic tested was mixed with 50 μL of another strain in microdilution plates in series. Later, 100 μL of a tested pathogenic strain (10^8^ CFU/mL) was added, and the plates were incubated at 37 °C for 48 h under suitable conditions. The fractional inhibition concentration (FIC) index of the probiotics in combination was calculated with the equation: FIC index = (MIC_A_, tested in combination)/(MIC_A_, tested alone) + (MIC_B_, tested in combination)/(MIC_B_, tested alone), where MIC_A_ and MIC_B_ were the MICs of the tested extract obtained between probiotics (*L. paracasei* SD1, *L. rhamnosus* SD4, and *L. rhamnosus* SD11). The combination was considered as synergistic, indifferent, and antagonistic when the FIC index was ≤0.5, >0.5 to <4, and >4, respectively.

### 2.4. Effect of SCFAs on Growth Suppression of Caco-2 and HIEC-6 Cells

#### 2.4.1. Cell Culture Preparation

A human colorectal adenocarcinoma cell line (Caco-2) and the human normal intestinal cell line (HIEC-6, ATCC^®^ CRL-3266™) were used in this study. The cells were cultured in Dulbecco’s modified eagle medium (DMEM), supplemented with 10% fetal bovine serum (*v*/*v*), 1% penicillin/streptomycin, 1% amphotericin B, and 1% non-essential amino acids, and were incubated at 37 °C under 5% CO_2_ for 3–5 days.

#### 2.4.2. Cell Suppression Assay

A monolayer of Caco-2 or HIEC-6 cells was seeded in 96-well plates at a concentration of 1 × 10^5^ cells/well and incubated at 37 °C under 5% CO_2_ for 3 days. Next, 100 µL of SCFA extract at various concentrations (2, 4, 6, and 8 mM) was added and incubated for different times (1, 2, 4, 6, and 24 h). Afterwards, the cells were washed twice with phosphate-buffered saline (PBS, pH 7.0). Growth suppression was determined using an MTT assay (3-(4,5-dimethylthiazol-2-yl)-2,5-diphenyltetrazolium bromide), and the untreated cells were used as a control. The percentage of growth suppression was calculated by the following formula: 100 − (OD of test sample × 100/OD of the control).

### 2.5. Effect of SCFAs on Antimicrobial Peptides and Anti-Inflammation

To determine the effect of the SCFAs produced by potential probiotics on antimicrobial peptides (hBD-2, hBD-3 and hBD-4) and anti-inflammatory agents (IL-10), the SCFA extracts of each individual probiotic strain were used at a final concentration of 5.5 mM (sub-lethal dose). Caco-2 and HIEC-6 cells at 1 × 10^5^ cells per well were seeded in 6-well plates and allowed to grow until the cells reached 90% confluency. The SCFA extracts from the individual strains tested were added at a final concentration of 5.5 mM and further incubated at 37 °C under 5% CO_2_ for 24 h. The hBD-2, hBD-3, hBD-4, and IL-10 mRNA expressions were examined using a real-time PCR (see below), and the untreated cells were used as a negative control.

### 2.6. Effect of SCFAs on Suppression of Pro-Inflammatory Cytokines

The Caco-2 and HIEC-6 cells were stimulated using cell walls of the individual pathogens to release pro-inflammatory cytokines, which were extracted using sonication and differential centrifugation, as described previously [32]. To determine the effect of the SCFAs produced by potential probiotics on the suppression of pro-inflammatory cytokines (IL-1β, IL-6, IL-8, and TNF-α), the Caco-2 and HIEC-6 cells were seeded in 6-well plates at 1 × 10^5^ cells per well and allowed to grow until they reached 90% confluency. The cells were exposed to cell walls extracted from each pathogen alone (final concentration of 100 µg/mL protein) or combined with the SCFAs of each probiotic strain alone (final concentration of 5.5 mM) or combined with the SCFAs of the probiotic strains in a combination. The exposure conditions were 37 °C and 5% CO_2_ for 24 h. Subsequently, the mRNA expression of IL-1β, IL-6, IL-8, and TNF-α were examined by real-time PCR. The untreated cells were used as a negative control.

#### The mRNA Expression by Real-Time PCR

Total RNA was extracted with the PureLink™ RNA Mini Kit and reverse transcribed using a RevertAid First Strand cDNA Synthesis Kit. The primers for the real-time PCR were as follows: hBD-2 (5′-CCAGCCATCAGCCATGAGGGT-3′ and 5′-GGAGCCCTTTCTGAATCCGCA-3′), hBD-3 (5′-AGCCTAGCAGCTATGAGGATC-3′ and 5′-CTTCGGCAGCATTTTCGGCCA-3′), hBD-4 (5′-TACACAGTTGCTGGGGATGA-3′ and 5′-GGTGCCAAGGACATCTAGGA-3′), IL-10: 5′-GCCTAACATGCTTCGAGATC-3′ and 5′-TGATGTCTGGGTCTTGGTTC-3′), IL-1β: (5′-CACGCTCCGGGACTCACAGC-3′ and 5′-CTGGCCGCCTTTGGTCCCTC-3′), IL-6: (5′-CGCCCCACACAGACAGCCAC-3′ and 5′-AGCTTCGTCAGCAGGCTGGC-3′), IL-8: (5′-TTTCTGATGGAGAGAGCTCTGTCTGG-3′ and 5′-AGTGGAACAAGACTTGTGGATCCTGG-3′), TNF-α: (5′-TTCTGCCTGCTGCACTTTGGA-3′ and 5′-TTGATGGCAGAGAGGAGGTTG-3′), and GAPDH (5′-ACCACAGTCCATGCCATCACTGC-3′ and 5′-TCCACCACCCTGTTGCTGTAGC-3′). Real-time PCR was performed for 40 cycles for all markers at the denaturing temperature at 95 °C for 20 s; different annealing temperatures for 20 s (60 °C for hBD-2, hBD-3, IL-1β, IL-6, IL-8, TNF-α, and GAPDH; 57 °C for IL-10; and 58 °C hBD-4); and the polymerizing temperature at 72 °C for 25 s.

Quantitative real-time PCR was performed using 5% cDNA (vol/vol) and the Sensi-fast™ SYBR No-ROX reagent (Bioline reagent Ltd., Memphis, TN, USA) with the CFX96 Touch™ Real-Time PCR detection system (Bio-Rad, Hercules, CA, USA). The relative induction of cytokine mRNA expression was normalized by the expression of GAPDH, which is the most stable expressed gene and one of the most commonly used housekeeping genes for comparing gene expression and showing the least amount of variability. The induction of cytokine mRNA expression by the SCFA extracts of the probiotics was compared with the mRNA expression induction of the cytokines by the untreated sample, the value for which was set to 1.0.

### 2.7. Statistical Analysis

The results were presented as the means ± standard deviation (SD). Statistical analysis was performed using the SPSS Statistics 22 software (IBM Analytics, Armonk, NY, USA). Comparisons between groups were analyzed using the Kruskal–Wallis test for non-parametric data. Differences between the two groups were analyzed using the Mann–Whitney test as appropriate. *p*-values of *p* < 0.05 were considered as statistically significant.

## 3. Results

### 3.1. Optimization of SCFA Production and Conditions

The SCFAs produced by individual probiotic strains in culture media were analyzed using the GC-MS. An example of the SCFAs of *L. rhamnosus* SD11 is demonstrated in Figure 1. It was shown that that SCFA peaks have a relatively low signal (Figure 1A). After the extraction of 10 mL of culture medium, an increase in the SCFA peaks of acetic acid (at 13.6073 min), propionic acid (at 15.1157 min), and butyric acid (at 16.3459 min) was clearly obtained, while the peak of the lactic acid was dramatically decreased (Figure 1B).

The butyrate level and the propionate level of each strain were evaluated with the calibration curve. This study focused on butyrate production; therefore, only the butyrate production of the strains tested was presented.

The probiotic growth and the butyric acid production of the individual strains at different temperatures (30, 37, 40, and 45 °C) and different times (0, 12, 24 and 48 h) are demonstrated in Figure 2. The probiotic growth and butyric acid gradually increased following the time of incubation. The greatest growth of all the probiotic strains was observed at 30 °C; there was an increase of approximately 5 LogCFU/mL at 48 h compared to the initial time of incubation. The highest butyric acid productions of all the strains were found at 45 °C, while the growths were low compared to the other temperatures used, ranging from approximately 1.0–1.7 LogCFU/mL during 12–48 h of incubation.

A ratio of butyric acid and growth is demonstrated in Figure 3. An increase in temperature resulted in an increase in butyrate production, and the maximum production was found at 45 °C in all the strains at each time point. The optimal period for the maximum butyrate production depended on the individual strains. At 12 h, *L. rhamnosus* SD11 showed the highest peak (25.3 mM, 2.2 g/L) compared to the other strains (Figure 3A); however, the maximum levels were reached at 24 h for *L. paracasei* SD1 (38.2 mM, 3.4 g/L) and *L. rhamnosus* SD11 (37.6 mM, 3.3 g/L) (Figure 3B). The maximum butyrate production for *L. rhamnosus* SD4 (28.9 mM, 2.5 g/L) and *L. rhamnosus* GG (26.4 mM, 2.3 g/L) was found at 48 h (Figure 3C). At 45 °C at 24 h, the highest butyrate production was observed with the combination of *L. paracasei* SD1 and *L. rhamnosus* SD11 (40.6 mM, 3.6 g/L), followed by *L. paracasei* SD1 alone (38.2 mM, 3.4 g/L) and *L. rhamnosus* SD11 alone (37.6 mM, 3.3 g/L) (Figure 3D).

### 3.2. Antimicrobial Activity of SCFAs Produced by Potential Probiotics against Pathogens

The antibacterial activity of the SCFAs extracted from the supernatant (cells incubated at 45 °C at 24 h) of four probiotic strains (*L. paracasei* SD1, *L. rhamnosus* SD4, *L. rhamnosus* SD11, and *L. rhamnosus* GG) against various pathogens were investigated using the agar diffusion method; the results are shown in Table 1. The results showed a difference in the inhibited zone, which ranged from 9 to 17 mm. Overall, *L. paracasei* SD1 and *L. rhamnosus* SD11 exhibited a significantly stronger antagonistic activity (a larger zone of inhibition) than the *L. rhamnosus* SD4 and LGG strains against all the tested pathogens. The SCFAs extracted from all four of the probiotic strains could inhibit the growth of *P. gingivalis* and *F. nucleatum* statistically significantly better than *S. enterica* and ETEC. Both *L. paracasei* SD1 and *L. rhamnosus* SD11 showed the significantly highest antibacterial activity against *P. gingivalis*, followed by *F. nucleatum*, ETEC and *S. enterica*.

Additionally, the minimum inhibitory concentration (MIC) of the extracted SCFAs against the four pathogens (*S. enterica*, ETEC, *F. nucleatum*, and *P. gingivalis*) was determined; the results concurred with the findings of the agar diffusion method (Table 1). The mean MICs of all the probiotic strains tested against *F. nucleatum* and *P. gingivalis* were lower than those against *S. enterica* and ETEC. Combinations of SCFA extracts from different probiotic strains showed synergistic effects, as indicated by their FIC indices, which were ≤0.5 (Table 1).

*L. paracasei* SD1 and *L. rhamnosus* SD11 individually showed the best antimicrobial effects against all the pathogens, and the combination of the probiotic strains correlated with the synergistic interaction. Thus, *L. paracasei* SD1 and *L. rhamnosus* SD11 in combination were selected for the next experiment.

### 3.3. Effect of SCFAs on Growth Suppression of Caco-2 and HIEC-6 Cells

We evaluated the effect of the SCFAs extracted from the lyophilized powders of the supernatants of the cells cultured at 45 °C on the suppression of the Caco-2 and HIEC-6 cells, using individual probiotic strains or a combination of the probiotic strains (Figure 4). The suppression effect of the SCFAs on the growth of the Caco-2 cells depended on the dose of the SCFAs and the time of exposure; a higher SCFA concentration and time of exposure resulted in a higher growth suppression of the cells treated. Figure 4A presents the growth suppression of the Caco-2 cells by 8 mM of butyrate extracted from the lyophilized powder of the supernatants of *L. paracasei* SD1 and *L. rhamnosus* SD11 cultured in combination for different periods.

In the Caco-2 cells, different probiotic strains exerted different levels of growth suppression. *L. paracasei* SD1 (94.1 ± 1.1%) and *L. rhamnosus* SD11 (93.1 ± 0.2%) induced a significantly higher growth suppression than *L. rhamnosus* SD4 (82.3 ± 0.4%) and *L. rhamnosus* GG (86.7 ± 0.5%). The combination of the *L. paracasei* SD1 and *L. rhamnosus* SD11 strains produced a significantly higher growth suppression (98.3 ± 0.1%) than the individual strains alone (Figure 4B). In HIEC-6 cells, the growth suppression was significantly lower than in the Caco-2 cells for all the probiotic strains. The probiotic strains showed no significant difference in the growth suppression of the HIEC-6 cells (Figure 4B).

### 3.4. Effect of SCFAs Produced by Potential Probiotics on Antimicrobial Peptide and Anti-Inflammation IL10 

Any effect of the SCFAs on the antimicrobial peptides and the cytokine suppression in the Caco-2 cells depended on the dose of SCFAs; the IC_50_ was estimated to be 5.5 mM for 24 h treatment with the combination of butyrate extracts from the *L. paracasei* SD1 and *L. rhamnosus* SD11 cultures. We investigated the effect of the SCFAs produced by the probiotics on the antimicrobial peptides, human β-defensin (hBD-2, hBD-3, and hBD-4) genes, and IL-10 expression in Caco-2 cells. The results showed that the levels of antimicrobial peptide mRNA expression were significantly different (Figure 5A). hBD-2 displayed a significantly higher expression than the others. L. rhamnosus GG (22.6 ± 0.7 folds) and a combination of L. paracasei SD1 and L. rhamnosus SD11 (21.4 ± 0.0 folds) produced the highest hBD-2 expression compared to the others.

In contrast, the hBD-2 expression in the HIEC-6 cells was significantly lower than in the Caco-2 cells (Figure 5B). However, the combination of *L. paracasei* SD1 and *L. rhamnosus* SD11 resulted in a significantly higher IL-10 expression in the HIEC-6 cells than in the Caco-2 cells (4.2 ± 0.2 folds).

### 3.5. Effect of SCFAs on Suppression of Pro-Inflammatory Cytokines

The suppression effect of the SCFAs produced by the probiotic strains on the pro-inflammatory cytokines IL-1β, IL-6, IL-8, and TNF-α was investigated in the Caco-2 cells (Figure 6). Among the pro-inflammatory cytokines tested, IL-8 showed significantly higher levels after stimulation with the cell walls of individual the pathogens, especially *P. gingivalis*. The SCFAs produced by the probiotic strains could reduce pro-inflammatory cytokine expression, which was clearly shown for the IL-8 expression, particularly with the combination of SCFAs produced by *L. paracasei* SD1 and *L. rhamnosus* SD11.

The SCFA-induced suppression of the pro-inflammatory cytokines was similar for the HIEC-6 cells and Caco-2 cells; however, the expressions were lower for the HIEC-6 cells than for the Caco-2 cells (Figure 7).

## 4. Discussion

SCFAs, particularly butyrate, have received considerable attention for their anticancer efficacy as they can delay or prevent CRC [15,16,33]. A number of studies have reported that certain *Lactobacillus* strains could produce butyrate, and in vivo studies have proven the anti-cancer effects of some of these strains [13,15,16,33]. To achieve the maximum desirable benefits of the products from the probiotic strains, the studies have searched for the optimal conditions needed [34,35,36]. One study showed that the optimal conditions in which *Lactobacillus acidophilus* CRL1259 inhibited uropathogenic *Escherischia coli* were pH 6.5 or 8.0 at 37 °C for 8h [36]. Another study reported that the optimal conditions for *Lactobacillus salivarius* in filmogenic solution were 45 °C for 4 h of fermentation [35]. This indicates that the individual optimal conditions were varied and depended on various factors, such as the type of probiotic strains, the temperatures, the times of incubation, the media of culture, etc.

This study attempted to carry out the optimal conditions for the butyric acid production of certain selected probiotic strains because there were limitations in the information that had already been reported. Most of the studies performed the experiments at 37 °C for 24 h, which was the temperature normally used [5,20]. No information that varied the temperatures and time of incubation has been reported. The initial findings in the present study revealed that butyrate production depended on the type of strain, the temperature, and the incubation period during cultivation. All the tested strains produced butyrate at different levels; however, *L. paracasei* SD1 and *L. rhamnosus* SD11 were stronger butyrate producers compared to the others. This observation is in agreement with previous studies which reported that butyrate production depends on the type and number of microorganisms [20,37]. For all the strains, 45 °C appeared to be the optimal temperature for maximum butyrate production; however, the difference in incubation time during cultivation was unique for individual strains. The butyrate levels found in this study ranged from 250 to 360 mg/L (when extracted directly from the supernatant rather than the lyophilized form), which were relatively higher than the previous reports showing butyrate levels of 2.89 mg/L for *Lactobacillus plantarum* O4T10E cultured at 37 °C [20] and 212.4 mg/L for *Clostridium butyricum* [5]. However, it is quite difficult to compare strains in different studies because various factors, such as the type of strain, the culture conditions, the incubation time, and the sample preparation could lead to different results. In addition, it should be mentioned here that the temperature for the butyrate yield seemed to be inversely related to the temperature of the probiotic growth. The results from this study provide crucial information because the properties and functions of the probiotic strains are strain-specific. Therefore, with the selection of strains and the conditions used for culturing, the desirable probiotics are essential for achieving maximal effectiveness.

Butyrate, together with the SCFAs (at 45 °C at 24 h), was further investigated for its ability to modulate gut microbiota. Our results showed that the butyrate produced by all the probiotic strains exhibited anti-microbial effects by inhibiting the growth of all the tested pathogens; however, differences were observed among the strains. A stronger antimicrobial activity was found with both *L. paracasei* SD1 and *L. rhamnosus* SD11 than with the other strains, especially against *P. gingivalis* and *F. nucleatum* (oral pathogens associated with periodontitis). Notably, synergistic activity was observed when combinations of the probiotic strains were tested. The current evidence indicates that *P. gingivalis* and *F. nucleatum* can promote the clinical and molecular features of CRC [8,38,39]. Accumulating evidence has shown that the antimicrobial activities of probiotic strains are derived from both antimicrobial protein substances and various acids [40,41]. SCFAs are expected to be partly involved in such activity, as demonstrated by the present study. Our results may support the findings from previous studies, which revealed that certain probiotic strains could modulate the gut microbiota by decreasing the levels of some pathogenic bacteria and increasing the levels of certain beneficial bacteria [42].

The previous studies suggested that butyrate-producing bacteria potentially exhibited anti-tumor properties in cancer cell cultures and animal models of cancer [43]. Concordantly, our results showed that the SCFAs extracted from the selected probiotic strains inhibited the growth of cancer cells (Caco-2); however, different strains show different activities. The growth-suppression effect of SCFAs on Caco-2 cells increased in a dose- and time-dependent manner, which is similar to the results from previous studies [44]. Among SCFAs, butyrate plays the most crucial role in intestinal physiology. It provides more than 70% of the energy needed to fuel the metabolic activity of coloncytes, which lower colorectal oncogenesis by modulating the signaling pathways to induce the apoptosis of cancer cells [43]. The mechanism by which butyrate inhibits cancer cell growth has been attributed to normal colonocytes using butyrate as their primary energy source. In contrast, cancerous colonocytes rely on glucose as a primary energy source; butyrate has been reported to accumulate and function as a histone deacetylase inhibitor, leading to the alteration of several important oncogenic signaling pathways [43]. The results of the present study are consistent with a previous study that reported the inhibition of cancerous proliferation by butyrate at a dose higher than 5 mM. The previous study revealed that the concentration of butyrate required to inhibit 50% Caco-2 cells (IC_50_) was 5.5 mM, which is close to that of a genetically engineered bio-butyrate produced by the probiotic *E. coli* Nissle 1917 (4.5 mM) [44]. A 5 mM dose of butyrate in vitro is physiologically relevant because this concentration has also been found to be effective in the lumen of the colon [45]. Similarly, at this concentration, butyrate was shown to inhibit cell proliferation in normal HIEC-6 cells, which concurred with the results of a study in FHC cells (a human normal colon cell) that showed a decrease in cell proliferation with 5 mM of butyrate [46]. However, the results from this study demonstrate that the inhibition effect of 8 mM of butyrate was greater with the Caco-2 cells (98.3 ± 0.1%) than with the HIEC-6 cells (78.1 ± 0.7%). This finding may be explained by the Warburg effect [46,47]. The cancer cells were dominantly directly affected with the 8 mM of butyrate level because butyrate accumulates in the nucleus of cancer cells. As such, it acts as an inhibitor of histone deacetylase (HDAC) and induces death in the CRC cells. In contrast, the normal cells usually use butyrate as the main energy source, resulting in a relatively low level of accumulated butyrate, producing a less inhibitory effect in the normal cells.

The immunomodulatory effect of the SCFAs produced by potential probiotics on antimicrobial peptides and IL-10 and the suppression of pro-inflammatory cytokines have been discussed together because of the relationship between these functions. Antimicrobial peptides, including hBD, have been recognized as essential antimicrobial defense molecules, and the decreased expression of these peptides could lead to chronic inflammation of the human intestinal mucosa [48]. A previous study showed that the human colon epithelial cells HT-29 and Caco-2 could constitutively express hBD-1 but not hBD-2. Abundant hBD-2 expression was found in the epithelium of an inflamed colon but not in the epithelium of a normal colon. This indicates that hBD-2 upregulated or induced a part of the inflammatory response [49]. Regarding the effects of probiotics on antimicrobial peptides, a study reported that *E. coli* Nissle 1917, a probiotic strain, could strongly induce the expression of hBD-2 in Caco-2 cells in a time- and dose-dependent manner [50].

Concordantly, our results revealed that the expression of hBD-2 was higher than that of hBD-3 and hBD-4, and certain probiotic strains, such as *L. rhamnosus* GG, as well as *L. paracasei* SD1 and *L. rhamnosus* SD11 in combination, produced a higher hBD-2 expression compared to the others. A similar result was observed for the IL-10 expression. All the hBDs, as well as the IL-10, were expressed at a lower level in HIEC-6 (normal human intestinal cells). The study suggested that the regulation of hBD expression is different among mucosal tissues [49], which needs to be further elucidated.

SCFAs are naturally produced by certain bacteria in the gut; however, the amount produced may not be sufficient to inhibit the development of CRC in certain circumstances. Previous studies have demonstrated that the level of butyrate-producing bacteria in the gut microbiota is lower in CRC patients compared to that in healthy subjects [51,52]. In addition, an inverse correlation has been found between fecal butyrate and tumor size in CRC [53]. A reduction of 1 µg/L of butyric acid in the feces is estimated to increase the risk of CRC development by 84.2% [54]. Butyrate has been receiving increasing attention for chemotherapy or chemoprevention because of its ability to inhibit tumor growth; however, butyrate as a chemotherapeutic agent has limitations. Butyrate can be rapidly taken up and metabolized by normal cells before reaching tumor cells, resulting in a half-life of 6 min and peak blood levels below 0.05 mM [48]. Moreover, the oral administration of butyrate is uncomfortable due to its rancid smell and unpleasant taste [55]. Some studies have shown that probiotic supplementation could increase SCFA concentration and provide health benefits to the host [5,56]. Therefore, certain probiotic strains may serve as potential natural resources for butyrate-producing strains and as candidates for the prevention of CRC. Although butyrate has been demonstrated to be an important acid for the function of the colonic epithelium and the immune and inflammatory responses, the butyrate levels produced from the probiotic strains were relatively low. As the SCFA is a complex of organic acids, which comprise major acetic, propionic, and butyric acids together with some other minor acids [13,14], the interaction among the organic acids may be the key to their role in maintaining health; this was not the function of any one acid alone.

In addition, the combination of probiotic strains exhibited stronger effects than one strain alone. Recently, it has been suggested that probiotic strains and species used in combination as multi-strain and multispecies probiotics have more advantages in comparison to single-strain probiotics [57]. For example, the combination of *Lactobacillus acidophilus* CGMCC 7282 with *Clostridium butyricum* CGMCC 7281 produced a stronger anti-inflammatory activity than either strain alone in mice [58]. Therefore, well-designed multispecies probiotics may provide benefits resulting from synergism when the probiotic effects of different probiotic strains are combined. In addition, to achieve the health benefits of probiotic strains in vivo, the other factors should be considered, especially probiotic survival in the gastrointestinal tract (GIT). The abilities to tolerate acid and bile salts of the probiotic strains used in the present study have been proven in previous reports [59]. However, some other factors, such as carrier matrices, sensory properties, shelf life, etc., should be considered [60] for the achievement of desirable health benefits.

## 5. Conclusions

In conclusion, the results from this study expand the knowledge regarding the ability of probiotic strains to produce butyrate, which can be altered by various factors, including the type of strain, the combination of strains, the temperature, and the incubation time for cultivation. *L. paracasei* SD1 and *L. rhamnosus* SD11, with the optimal conditions at 45 °C for 24 h, provided the greatest amount of butyrate compared to the others. Collectively, the results of this study demonstrate that the selected probiotic strains exert beneficial anti-carcinogenesis effects through SCFAs, the including inhibition of pathogen-related CRC, the stimulation of antimicrobial peptides, the inhibition of cancer cells, and anti-inflammation. Our results indicate that certain probiotic strains, particularly *L. paracasei* SD1 and *L. rhamnosus* SD11 in combination, may serve as a good natural resource for SCFAs. However, to evaluate the relationship between the butyrate production of the selected probiotic strains and their impact on the prevention of CRC in the improvement of human health, further clinical trials on its use as a potential anticancer therapeutic are needed.

## Figures and Tables

**Figure 1 biomolecules-12-01829-f001:**
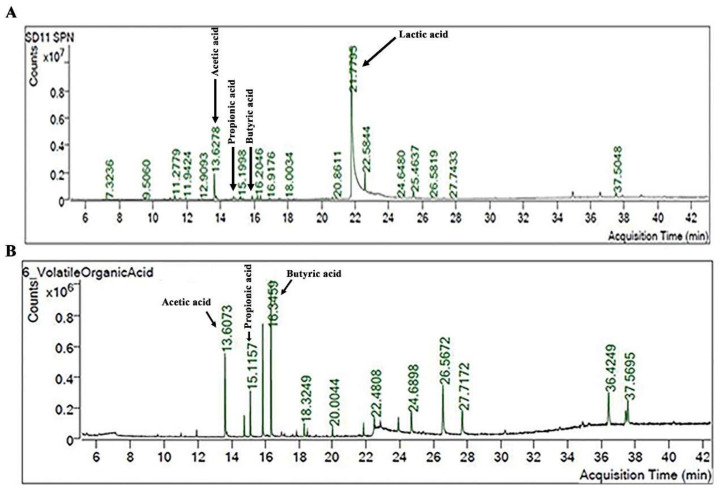
The GC-MS profile of SCFAs produced by *L. rhamnosus* SD11 before (**A**) and after extraction (**B**).

**Figure 2 biomolecules-12-01829-f002:**
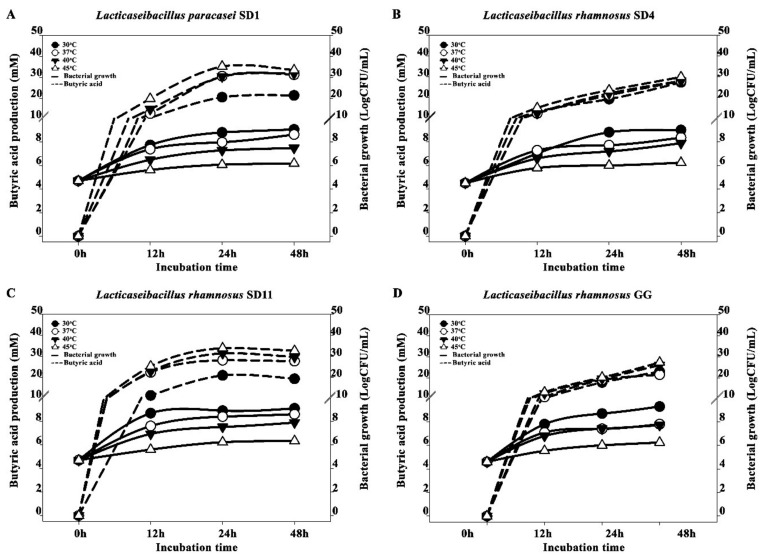
Butyric acid production and bacterial growth at different incubation times of various probiotic strains; (**A**) *L. paracasei* SD1, (**B**) *L. rhamnosus* SD4, (**C**) *L. rhamnosus* SD11, and (**D**) *L. rhamnosus* GG.

**Figure 3 biomolecules-12-01829-f003:**
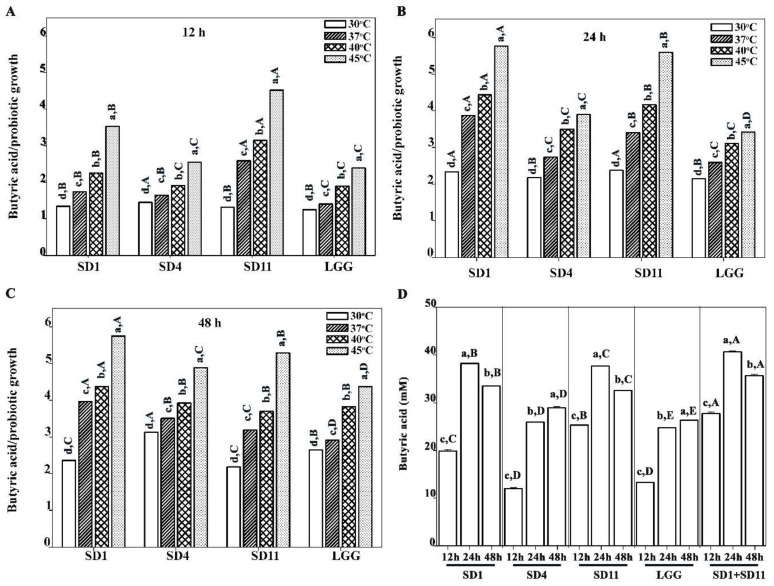
Ratio of butyric acid production and bacterial growth rate at different incubation times: (**A**) 12 h, (**B**) 24 h, and (**C**) 48 h; (**D**) butyric acid production (mM) by probiotics cultured at 45 °C at various incubation times of 12, 24, and 48 h. Lowercase letters show significant differences between different incubation temperatures (**A**–**C**) or significant differences between different incubation times (**D**) at each probiotic strain (*p* < 0.05). Uppercase letters show significant differences between different probiotic strains (**A**–**C**) at each incubation temperature or show significant differences between different probiotic strains at each incubation time (**D**) (*p* < 0.05). Statistical analysis was performed with the Kruskal–Wallis test, followed by the Mann–Whitney U test.

**Figure 4 biomolecules-12-01829-f004:**
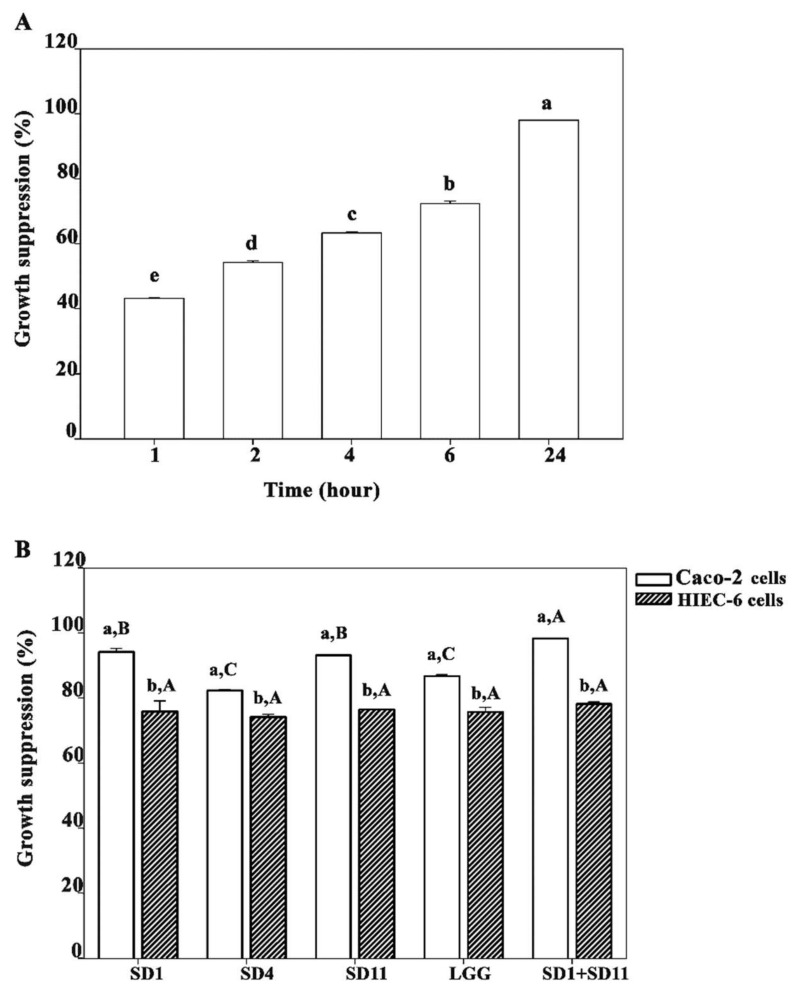
The growth-suppression percentage for 8 mM butyrate produced from probiotics: (**A**) time-dependent growth suppression of Caco-2 cells and (**B**) growth suppression of Caco-2 and HIEC-6 cells by individual probiotic strains and a combination of *L. paracasei* SD1 and *L. rhamnosus* SD11 at 24 h. Lowercase letters indicate (**A**) statistical difference in growth suppression by short-chain fatty acids (SCFAs) at various times or (**B**) statistical difference in growth suppression of Caco-2 or HIEC-6 cells by individual probiotic strains (*p* < 0.05). Uppercase letters indicate statistical difference in growth suppression of either Caco-2 cells or HIEC-6 cells treated with SCFAs produced from each probiotic strain and a combination of *L. paracasei* SD1 and *L. rhamnosus* SD11 (*p* < 0.05). The statistical analysis was performed using the Kruskal–Wallis test followed by the Mann–Whitney U test.

**Figure 5 biomolecules-12-01829-f005:**
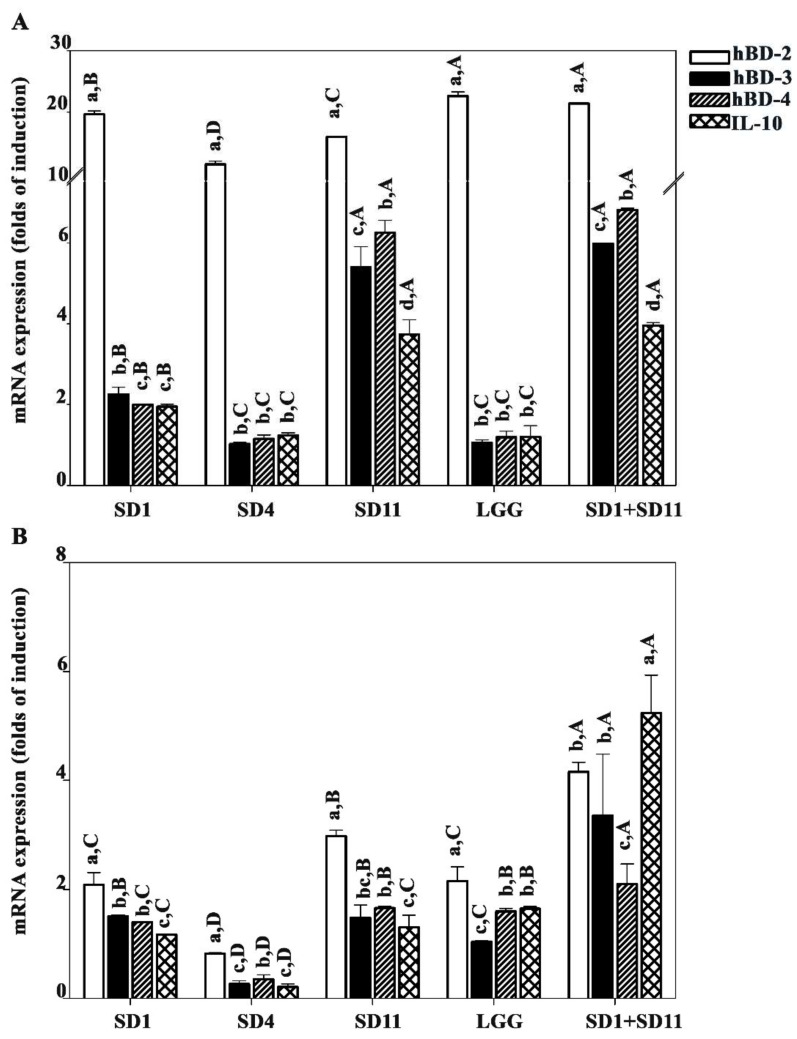
Antimicrobial peptides and anti-inflammatory cytokine (IL-10) stimulated by SCFAs from the probiotic strains in (**A**) Caco-2 and (**B**) HIEC-6 cells, respectively. Lowercase letters indicate significant differences between mRNA expression parameters for each probiotic strain (*p* < 0.05). Uppercase letters indicate significant differences between different probiotic strains with respect to each expression parameter (antimicrobial peptides or anti-inflammatory cytokine) (*p* < 0.05). Statistical analysis was performed using the Kruskal–Wallis test followed by the Mann–Whitney U test.

**Figure 6 biomolecules-12-01829-f006:**
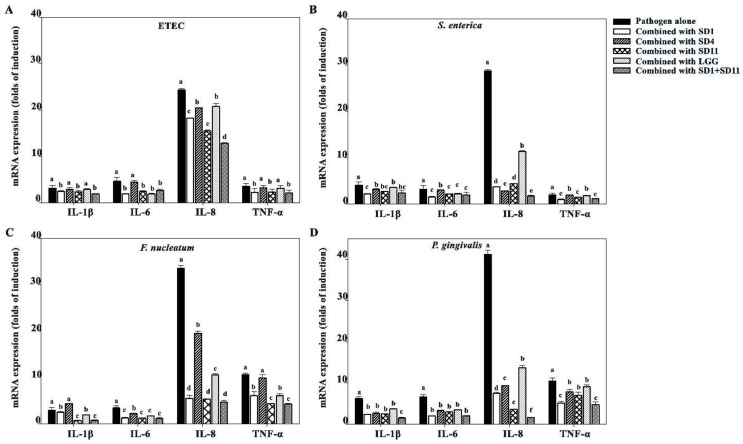
Effect of SCFAs on suppression of pro-inflammatory cytokines produced upon stimulation with various pathogen strains in Caco-2 cells; (**A**) ETEC, (**B**) *S. enterica*, (**C**) *F. nucleatum*, (**D**) *P. gingivalis*. Lowercase letters represent significant differences in the expression of each cytokine upon stimulation with the pathogen alone and upon exposure to the pathogen combined with probiotic strains (*p* < 0.05). Statistical analysis was performed using the Kruskal–Wallis test followed by the Mann–Whitney U test.

**Figure 7 biomolecules-12-01829-f007:**
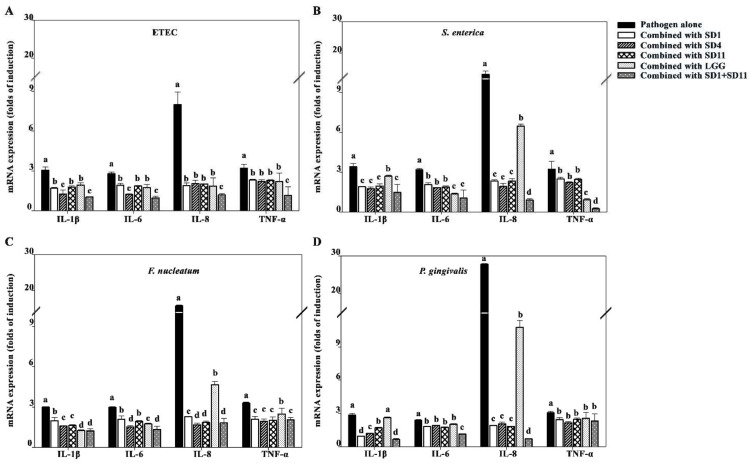
Effects of SCFAs on suppression of pro-inflammatory cytokines stimulated with various pathogen strains in HIEC-6 cells; (**A**) ETEC, (**B**) *S. enterica*, (**C**) *F. nucleatum*, (**D**) *P. gingivalis*. Lowercase letters represent significant differences in the expression of each cytokine upon stimulation with pathogen alone and upon exposure to pathogen combined with probiotic strains (*p* < 0.05). The statistical analysis was performed using the Kruskal–Wallis test followed by the Mann–Whitney U test.

**Table 1 biomolecules-12-01829-t001:** Antimicrobial activity of short-chain fatty acids (SCFAs) of probiotics against pathogens.

Probiotics	*S* *. enterica*	ETEC	*F* *. nucleatum*	*P* *. gingivalis*
Antimicrobial activity by agar diffusion method (mean ± SD in mm)				
SD1	12.5 ± 0.0 ^c,B^	12.5 ± 0.4 ^c,A^	15.0 ± 0.4 ^b,A^	17.0 ± 0.4 ^a,A^
SD4	9.0 ± 0.0 ^c,C^	10.0 ± 1.1 ^c,B^	12.0 ± 0.4 ^b,C^	14.0 ± 0.0 ^a,C^
SD11	12.3 ± 0.0 ^c,A^	12.0 ± 0.4 ^c,A^	15.0 ± 0.7 ^b,A^	17.0 ± 0.4 ^a,A^
LGG	10.6 ± 1.1 ^c,B^	10.0 ± 0.0 ^c,B^	13.0 ± 0.4 ^b,B^	15.5 ± 0.4 ^a,B^
MIC by broth microdilution method (mean ± SD in µg/mL)				
SD1	517.3 ± 0.0 ^a,B^	517.3 ± 0.0 ^a,B^	358.6 ± 0.0 ^b,B^	358.6 ± 0.0 ^b,B^
SD4	537.6 ± 0.0 ^a,A^	537.6 ± 0.0 ^a,A^	368.8 ± 0.0 ^b,A^	368.8 ± 0.0 ^b,A^
SD11	517.3 ± 0.0 ^a,B^	517.3 ± 0.0 ^a,B^	358.6 ± 0.0 ^b,B^	358.6 ± 0.0 ^b,B^
FIC index				
*L*. *paracasei* SD1 + *L*. *rhamnosus* SD4	1.0 (258.7/268.8)	1.0 (258.7/268.8)	0.5 (89.7/92.2)	0.5 (89.7/92.2)
*L*. *paracasei* SD1+ *L*. *rhamnosus* SD11	1.0 (258.7/258.7)	1.0 (258.7/258.7)	0.5 (89.7/89.7)	0.5 (89.7/89.7)
*L*. *rhamnosus* SD4+ *L*. *rhamnosus* SD11	1.0 (268.8/258.7)	1.0 (268.8/258.7)	0.5 (92.2/89.7)	0.5 (92.2/89.7)

FIC index ≤ 0.5, synergy; FIC index > 0.5 to 4, indifference; FIC index > 4, antagonism. Lowercase letters indicate a significant difference in the same row (*p* < 0.05), as per the Kruskal–Wallis test followed by the Mann–Whitney U test. Uppercase letters indicate a significant difference in the same column (*p* < 0.05), as per the Kruskal–Wallis test followed by the Mann–Whitney U test.

## Data Availability

The data presented in this study are available on request from the corresponding author.

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
