# Peer review of "Characterization of Short Chain Fatty Acids Produced by Selected Potential Probiotic Lactobacillus Strains"

_biomolecules, 2022, doi:10.3390/biom12121829_

Round 1

Reviewer 1 Report (Previous Reviewer 2)

Th current version is adequate for publication.

I have one comment: Lines 393-397: The authors discuss the fact that the effects of probiotics are often strain-specific, which is indeed important to mention. To make this section more complete, I would like the authors also to briefly discuss the fact that probiotics efficacy is ultimately also determined by all kind of final product parameters, like the matrix. Please refer for this part of the discussion to:

Flach, J., van der Waal, M. B., van den Nieuwboer, M., Claassen, E., & Larsen, O. F. A. (2018). The underexposed role of food matrices in probiotic products: Reviewing the relationship between carrier matrices and product parameters. Critical reviews in food science and nutrition, 58(15), 2570-2584.

Author Response

2nd December 2022

Ref: Manuscript ID 2011747

Title: Characterization of short-chain fatty acids produced by selected potential probiotic strains.

Dear Editor,

We are very grateful for your kind suggestions and those of the referees. We found them to be very useful and have carefully revised and improved the manuscript considering all the referees’ suggestions. The changes have been marked in yellow highlight.

Yours sincerely,

Prof. Dr. Rawee Teanpaisan

Department of Oral Diagnostic Sciences

Faculty of Dentistry

Prince of Songkla University

Reviewer 2 Report (New Reviewer)

This scientific article communicated by Thananimit et al. systemically performed a series of experiments to examine both SCFAs-producing ability of the selected probiotic strains and functions of SCFA. Their great energy to investigate the issue should be evaluated and is thought be worth publishing. However, there are some problems that are unable to ignore in the study.

1.    As authors also demonstrated in Figure 1A, it has been well known that lactobacilli dominantly produce lactic acid but only a little SCFAs including butyric acid. In their study, however, the lactobacillus strains were adopted as the source of SCFAs-producing bacteria. Why didn’t they use probiotic bifidobacteria that produce a lot of SCFAs or butyrate-producing bacteria such as Clostridium butyricum in their study ?

2.    In the DISCUSSION, it was quoted that about 5 mM of butyrate is necessary to significantly suppress the carcinogenesis of CRC in the gut. Because the amounts of SCFAs secreted by naturally growing lactobacilli to the supernatant was far lower than that, it was suspected that authors concentrated the supernatants to extract biologically effective amount of SCFAs. Are the results using such artificial treatment able to translate to the biological use of lactobacillus strains as probiotics? 

3.    According to Figure 1B, the lyophilized powder made of supernatants appears to include some un-identified molecules other than SCFAs. Is there any possibility that those unknown molecules are also involved in the induction of the biological effects? Comparison between the biological effects exerted by the lyophilized powder and the standard chemical samples of SCFAs may solve the query.

4.    If the author’s final goal were to screen probiotic lactobacillus strains to prevent CRC, one of the strategies will be suppressing Fusobacterium nucleatum and Porphyromonas gingivalis inhabiting the gut as the authors quoted in INTRODUCTION. Because those two bacteria are Gram-negative species, they are highly sensitive to organic acids such as lactic acid. Therefore, lactobacillus strains, which are resistant to gastric acid and bile acid, and bear ability to have close proximity to intestinal mucosa, will be successful candidate probiotic strains for the prevention of CRC.

5.    The title should include the word of “Lactobacillus”

Author Response

2nd December 2022

Ref: Manuscript ID 2011747

Title: Characterization of short-chain fatty acids produced by selected potential probiotic strains.

Dear Editor,

We are very grateful for your kind suggestions and those of the referees. We found them to be very useful and have carefully revised and improved the manuscript considering all the referees’ suggestions. The changes have been marked in yellow highlight.

Yours sincerely,

Prof. Dr. Rawee Teanpaisan

Department of Oral Diagnostic Sciences

Faculty of Dentistry

Prince of Songkla University

Round 2

Reviewer 2 Report (New Reviewer)

No comment.  The aim of scientific reserch is far different between the authors and the reviewer.

This manuscript is a resubmission of an earlier submission. The following is a list of the peer review reports and author responses from that submission.

Round 1

Reviewer 1 Report

Characterization of short chain fatty acids produced by selected potential probiotic strains has been submitted to biomolecules (1909605) in special issue with title Non Herbal Nutraceutical, Probiotic, Vitamins and Fatty Acids in Cancer.

General formatting changes: add line numbers through the manuscript.

Abstract: Use same abbreviation of one word in the whole manuscript to avoid any confusion.

Keywords: it is better to add Short-chain fatty acids in keywords

Material and methods: why author has chosen 12 hours for growth period, provide any references for probiotics growth 12 can be used, explain

Explain source of each bacterial and how did to make sure these are pure and true as mentioned in manuscript.

Results

Use mL instead of ml throughout the manuscript

Compare your results for SD 11 @ 12 h and 45 °C with some reference papers because it is difficult to have such a higher growth in short time. Figure 2 (A)

Discussion:

In this part compare your findings with previous reported work and explain why your findings are different with scientific reasoning.

Conclusion

Add some conclusive data in this part the optimized growth conditions and revise it thoroughly to improve its readability.

Author Response

Comments to the Author

Characterization of short chain fatty acids produced by selected potential probiotic strains has been submitted to biomolecules (1909605) in special issue with title Non Herbal Nutraceutical, Probiotic, Vitamins and Fatty Acids in Cancer.

(1) General formatting changes: add line numbers through the manuscript.

Response: The line numbers has been added, as suggested.

(2) Abstract: Use same abbreviation of one word in the whole manuscript to avoid any confusion.

Response: All abbreviation was corrected in the manuscript.

Keywords: it is better to add Short-chain fatty acids in keywords

Response: The short-chain fatty acids has been included in the keywords.

(3) Material and methods: why author has chosen 12 hours for growth period, provide any references for probiotics growth 12 can be used, explain

Response: The design experiment for probiotics culture was to find the optimum conditions at various temperatures and incubation times to obtain the high yield of SCFAs not for the probiotic growth. Therefore, the various times have been set. The information has been added in lines 89-92, page 2.

(4) Explain source of each bacterial and how did to make sure these are pure and true as mentioned in manuscript.

Response: The details requested have been added in lines 62-65, page 2.

(5) Results

Use mL instead of ml throughout the manuscript

Response: The mL has been corrected in the manuscript.

Compare your results for SD 11 @ 12 h and 45 °C with some reference papers because it is difficult to have such a higher growth in short time. Figure 2 (A)

Response: This work was focused on the SCFAs production. The purpose of this work is to find the optimum conditions for probiotics culture consequently the maximum SCFAs product as mentioned in reply of question (3).

(6) Discussion:

In this part compare your findings with previous reported work and explain why your findings are different with scientific reasoning.

Response: The reasons have been added in lines 423-430, page 13.

(7) Conclusion

Add some conclusive data in this part the optimized growth conditions and revise it thoroughly to improve its readability.

Response: As suggested, the detail of optimized growth conditions has been added in section; conclusion in lines 481-482, page 16.

Reviewer 2 Report

The butyrate producing potential of probiotic strains is an important topic, and as such this paper is for sure of relevance. I do have, however two major concerns:

(1) The potential of the probiotic strains are studied as a function of temperature, and some experiments are performed at 45 degrees Celsius. As the body temperature is 37 degrees Celsius,  the experiments not performed at 37 degrees Celsius do not have direct in vivo / clinical relevance.

(2) All results are obtained from in vitro experiments. It is known that the properties of probiotics are dependent on the matrix they reside in (please refer to: Flach, J., van der Waal, M. B., van den Nieuwboer, M., Claassen, E., & Larsen, O. F. (2018). The underexposed role of food matrices in probiotic products: Reviewing the relationship between carrier matrices and product parameters. Critical reviews in food science and nutrition, 58(15), 2570-2584.).

The authors should thoroughly discuss how their results can be extrapolated and are of relevance to the in vivo human situation, regarding both body temperature and the in vivo "matrix".

Author Response

Comments to the Author

The butyrate producing potential of probiotic strains is an important topic, and as such this paper is for sure of relevance. I do have, however two major concerns:

(1) The potential of the probiotic strains are studied as a function of temperature, and some experiments are performed at 45 degrees Celsius. As the body temperature is 37 degrees Celsius, the experiments not performed at 37 degrees Celsius do not have direct in vivo / clinical relevance.

Response: The main purpose of this study is to obtain the optimum condition for receiving the maximum SCFAs production of probiotic strains (The details have been added in lines 89-92, page 2). The result showed that 45°C appears to be the optimal temperature for highest SCFAs production. However, the experimental in vivo for the effectiveness of SCFAs in prevention of the CRC has been planned for clinical trial (The information have been included in lines 472-476 on page 14.

(2) All results are obtained from in vitro experiments. It is known that the properties of probiotics are dependent on the matrix they reside in (please refer to: Flach, J., van der Waal, M. B., van den Nieuwboer, M., Claassen, E., & Larsen, O. F. (2018). The underexposed role of food matrices in probiotic products: Reviewing the relationship between carrier matrices and product parameters. Critical reviews in food science and nutrition58(15), 2570-2584.). The authors should thoroughly discuss how their results can be extrapolated and are of relevance to the in vivo human situation, regarding both body temperature and the in vivo "matrix".

Response: The explanation was added in lines 472-476, page 14.

Round 2

Reviewer 1 Report

The author does not comply with the comments. So this manuscript cannot be accepted in this form.